# Assessment of Preoperative TSH Serum Level and Thyroid Cancer Occurrence in Patients with AUS/FLUS Thyroid Nodule Diagnosis

**DOI:** 10.3390/biomedicines10081916

**Published:** 2022-08-08

**Authors:** Krzysztof Kaliszewski, Dorota Diakowska, Marta Rzeszutko, Łukasz Nowak, Beata Wojtczak, Krzysztof Sutkowski, Maksymilian Ludwig, Bartłomiej Ludwig, Agnieszka Mikuła, Maria Greniuk, Urszula Tokarczyk, Jerzy Rudnicki

**Affiliations:** 1Department of General, Minimally Invasive and Endocrine Surgery, Wroclaw Medical University, Borowska Street 213, 50-556 Wroclaw, Poland; 2Department of Basic Science, Faculty of Health Science, Wroclaw Medical University, Bartel Street 5, 51-618 Wroclaw, Poland; 3Department of Pathomorphology, Wroclaw Medical University, Marcinkowski Street 1, 50-368 Wroclaw, Poland; 4Department of Urology and Urological Oncology, Wroclaw Medical University, Borowska Street 213, 50-556 Wroclaw, Poland

**Keywords:** thyroid-stimulating hormone, TSH, atypia of undetermined significance, follicular lesion of undetermined significance, AUS/FLUS, thyroid cancer

## Abstract

Thyroid-stimulating hormone (TSH) is a growth factor associated with the initiation and progression of well-differentiated thyroid cancer (WDTC). Atypia of undetermined significance and follicular lesion of undetermined significance (AUS/FLUS) are the most uncertain cytological diagnoses of thyroid nodules. The aim of the study was to determine the association of histopathological diagnosis with preoperative serum TSH levels in patients with AUS/FLUS thyroid nodule diagnosis. Among 5028 individuals with thyroid nodules, 342 (6.8%) with AUS/FLUS diagnoses were analyzed. The frequency of all histopathology diagnoses was assessed for associations with preoperative serum TSH levels. The median TSH concentration was significantly higher in patients with AUS/FLUS diagnosis and histopathology of WDTC than in patients with the same cytology result and histopathology of a benign tumor (*p* < 0.0001). The diagnostic potential of serum TSH level was determined to evaluate risk of malignancy in patients with thyroid nodules classified into the Bethesda III category. ROC analysis showed the TSH concentration at a cutoff point of 2.5 mIU/L to be an acceptable prognostic factor for WDTC. For this optimal cutoff point, the AUC was 0.877, the sensitivity was 0.830, and the specificity was 0.902. Preoperative serum TSH levels in patients with AUS/FLUS thyroid tumor diagnosis should be taken into consideration in the decision-making process and clinical management.

## 1. Introduction

Clinical use of cytological diagnoses, such as atypia of undetermined significance (AUS) or follicular lesions of undetermined significance (FLUS), varies [1]; thus, management of these indeterminate groups, known as the “gray zone” category, is still controversial [2,3,4,5]. Such diagnoses are for thyroid nodules containing follicular lesions without nuclear atypia, known as FLUS, and for lesions with nuclear atypia included in follicular or nonfollicular patterns, i.e., AUS [5,6]. The risk of malignancy (ROM) of AUS/FLUS categories are treated together as one, and a third category (Bethesda III) is estimated within a wide range and makes up 10 to 30% according to the second, updated version of The Bethesda System for Reporting Thyroid Cytopathology (TBSRTC) introduced in 2018 [4,7]. According to the 2nd Edition of TBSRTC, the ROM of AUS/FLUS nodules increases to 30% if malignant lesions are assigned as noninvasive follicular thyroid neoplasm with papillary-like nuclear features (NIFTP), and accounts for 6–18% when NIFTP is excluded. According to the 1st TBSRTC Edition, it was previously estimated to be 5–15% [8]. Some authors have suggested that ROM with the AUS/FLUS category ranges from 25% to even 50% [7,9]. Hence, the majority of thyroid nodules assigned to this category qualify for diagnostic surgery to determine the final diagnosis [3,4,10]. Overall, there is considerable variability between cytopathologists as well as the institutions that use this category [11,12]. Based on Mosca et al.’s [6] opinion, we also confirmed that the value of this cytology diagnosis obtained through ultrasound-guided fine-needle aspiration biopsy (UG-FNAB) procedures is strongly dependent on the examiner’s experience and other technical details.

The emergence and the U.S. Food and Drug Administration (FDA) approval of molecular testing suggests that this percentage and its associated recommendation require re-examination [2]. Therefore, the discovery of biomarkers for predicting malignancy in AUS/FLUS thyroid nodules has become an urgent clinical need.

Thyroid-stimulating hormone (TSH) is secreted by the pituitary gland and promotes and stimulates the growth and function of the thyroid gland. Some authors suggest that TSH is a growth factor connected with the initiation and progression of well-differentiated thyroid cancer (WDTC) [13,14]. Patients with thyroid cancer (TC) reportedly have higher blood TSH levels than individuals with benign tumors [15]; others suggest that the TSH concentration can be used as a marker for the prediction of TC [16]. A higher concentration of TSH may predict thyroid malignancy, even if it is within the normal range [17]. In one of our previous studies, we estimated a significantly lower rate of TC in patients with category IV TBSRTC, in whom nonsuppressive thyroid hormone therapy was applied [18]. Category IV is also treated as the uncertain category, and many authors treat cases in this category individually [19]. We suggest that the mean serum TSH level is low in patients who undergo nonsuppressive thyroid hormone therapy [18]. We addressed whether nonsuppressive thyroid hormone therapy significantly decreases the rate of malignancy in category IV but not in category III. However, in the current study, we included only patients with category III (AUS/FLUS) disease with endogenous levothyroxine and not on nonsuppressive thyroid hormone therapy; cases of the AUS/FLUS category with nonsuppressive thyroid hormone therapy were excluded.

Some others have highlighted the clinical observation of higher TSH levels in patients with a more advanced stage of TC, which may suggest its role in the development and progression of malignancy [20].

We performed this retrospective analysis because of many suggestions supporting the argument that elevated blood TSH concentration might be connected with a higher ROM of thyroid nodules and because of a lack of evidence for a connection between cytology results and TSH level. We decided to analyze the most controversial group of the 2nd Edition of TBSRTC to help clinicians manage thyroid nodules assigned to the AUS/FLUS category.

The aim of this study was to determine the association of histopathological diagnosis with preoperative serum TSH levels among patients with AUS/FLUS thyroid nodule diagnosis. To the best of our knowledge, this is the first study conducted over a long duration (11 years) to identify these clinical observations in our country (Poland).

## 2. Materials and Methods

### 2.1. Study Population

We retrospectively analyzed 5024 medical records for patients admitted and surgically treated in the 1st Department and Clinic of General, Gastrointestinal and Endocrine Surgery due to TNs between January 2008 and December 2018 [21].

The study protocol was approved by the Institutional Review Board and Ethics Committee of Wroclaw Medical University, Wroclaw, Poland (No: KB 783/2017). All included patients provided admission informed consent, which stipulated that the results may be used for research purposes. The data were analyzed retrospectively and anonymously from established medical records. The authors did not have access to identifying patient information or direct access to the study participants.

### 2.2. Data Collection

From the initial group of patients (*n* = 5028), we extracted data for 342 (6.8%) patients with an AUS/FLUS diagnosis (category III according to the 2nd edition of TBSRTC). We excluded from the study patients who were under nonsuppressive thyroid hormone therapy, patients with a history of head and neck radiation exposure, individuals in a hyper- or hypothyroidism status, and patients with a history of family TC and previous radioactive iodine treatment or thyroid surgery.

In each of the medical records of these individuals, we evaluated the numbers and diagnoses of all biopsies performed during the observational study time. Using the patients’ medical records, data for age, sex, TSH level, cytological features and histological diagnosis were obtained. All patients underwent UG-FNAB of the thyroid nodule at least once before surgery, and the cytopathological slides were evaluated according to the 2nd Edition of TBSRTC. Additionally, we correlated the TSH level of the selected patients with previously analyzed ultrasound features of the thyroid nodules assigned an AUS/FLUS diagnosis. All procedures (ultrasonography, UG-FNAB, surgery) were performed by one experienced in thyroid diagnosis and treatment clinical team consisting of two pathologists, one radiologist, two oncologists, one endocrinologist and four surgeons. The selection of the study group is presented as a flow diagram (Figure 1).

### 2.3. Thyroid-Stimulating Hormone (TSH) Levels

We assessed the serum TSH level in all individuals enrolled in the analysis. The level was measured before surgery using an electrochemiluminescence immunoassay technique (Roche Corporation, Indianapolis, IN, USA). The normal range of TSH level in this study was 0.4–4.5 mIU/L.

### 2.4. The Bethesda System for Reporting Thyroid Cytopathology (TBSRT) 2nd Edition

All patients underwent ultrasound examination and UG-FNAB performed by one endocrine clinician team experienced in thyroid diseases. All cytopathological slides were assessed by two pathologists experienced in thyroid diseases according to the 2nd updated TBSRTC classification [7]. All of the specimens obtained from UG-FNAB of the thyroid nodules in patients operated on in 2008–2017 (study period: 2008–2018) were again reanalyzed and assigned to appropriate categories according to 2nd Edition of TBSRTC because this classification was formed, recommended, introduced, and used in 2018 [7].

### 2.5. Statistical Analysis

All calculations were performed using Statistica v. 13.3 (Tibco Software Inc., Palo Alto, CA, USA). The number of observations and percentages are used to describe qualitative data; the mean and standard deviation (±SD) or median and interquartile range (Q1-Q3) are used for presenting quantitative data. The distribution of continuous data was analyzed by the Kolmogorov–Smirnov test. The chi-square test, Fisher’s exact test, Student’s *t* test (for normally distributed quantitative data) and the Mann–Whitney test (for asymmetric distribution of quantitative data) were performed for comparative analyses between two independent groups.

The diagnostic potential of TSH concentration was determined by receiver operating characteristic (ROC) analysis with calculation of the cutoff point. The area under the ROC curve (AUC), sensitivity, specificity, accuracy, positive predictive value (PPV), negative predictive value (NPV), and Youden index were also calculated.

The stepwise method of univariable and multivariable logistic regression analyses was used for the determination of predictors of thyroid cancer presence. Odds ratios (ORs) and 95% confidence intervals (±95% CI) were also calculated. A two-tailed *p* < 0.05 was considered to be statistically significant.

## 3. Results

The total group of patients with thyroid nodules classified as Bethesda category III according to the 2nd Edition TBSRTC (*n* = 342) was divided into two subgroups: patients with benign thyroid tumors (*n* = 295) and patients with thyroid cancer (*n* = 47). Demographic, clinical, and histopathological characteristics of patients with malignant tumor are presented in Table 1.

Papillary thyroid cancer (PTC) was diagnosed in 13.4% (*n* = 46) of all AUS/FLUS patients, follicular thyroid cancer (FTC) in 0.3% (*n* = 1), adenoma in 15.8% (*n* = 54), thyroiditis in 12.9% (*n* = 44), and multinodular goiter in 55.8% (*n* = 197). The results of comparative analyses of demographic, clinical and ultrasound parameters in selected subgroups are presented in Table 2.

The number of female patients with thyroid malignancy was significantly higher than that of male patients (*p* = 0.003). Regarding nodules, TC was more often diagnosed in younger patients, i.e., below 55 years old (*p* < 0.05). We observed a significantly higher frequency of the occurrence of ultrasound features such as microcalcifications, hypoechogenicity, high vascularity, presence of irregular margins, and taller-than-wide tumors in patients with cancer (*p* < 0.0001 for all). The median TSH concentration was significantly higher in the cancer group than in the benign thyroid tumor group (*p* < 0.0001). Therefore, the diagnostic potential of serum TSH level was determined to evaluate the risk of malignancy in patients with thyroid nodules classified as the Bethesda III category. ROC analysis showed that TSH concentration at a cutoff point of 2.5 mIU/L may be an acceptable prognostic factor for thyroid cancer (Figure 2).

For this optimal cutoff point, the AUC was 0.877, the sensitivity was 0.830, and the specificity was 0.902 (Table 3).

The importance of TSH levels, demographic parameters, and ultrasound features as independent prognostic factors of thyroid malignancy was tested by univariable and multivariable logistic regression analyses (Table 4).

In univariable analysis, all independent variables were significant predictors of cancer, but the model constructed for multivariable analysis showed that TSH level above 2.5 mIU/L, microcalcifications, hypoechogenicity, taller-than-wide nodules, and age below 55 years old were leading indicators of thyroid malignancy.

## 4. Discussion

TBSRTC is the most accurate and cost-effective method for evaluating thyroid nodule diagnosis, but the AUS/FLUS category remains controversial and widely debated [2,3,4,5]. Although there are some studies that have sought to help clinicians manage patients with this diagnosis, they have mainly taken into consideration the imaging of tumors.

TSH secreted by the pituitary gland is a main factor regulating thyroid hormone production, thyroid function and, thus, thyroid disease [22]. Recently, it was highlighted that TSH also plays a role in the development and progression of TC by stimulating cell growth and proliferation [23]. TC cells have receptors for TSH on their membranes, stimulating growth and pathophysiological processes when activated [24,25]. Gudmundsson et al. [26] presented a negative association between TSH level and TC risk in a genome-wide association study, showing that the two most common variants of genes located at 9q22.23 and 14q13.3 are associated with low TSH levels. In another study, the same authors reported that a low TSH level is associated with a further TC risk linked to three variants located at 2q35, 8p12, and 14q13.3, demonstrating that a low TSH level might lead to less differentiation of the thyroid epithelium, with a possible higher susceptibility to malignant cell transformation [27].

Because TSH is recognized as a factor in the initiation, growth, and progression of well-differentiated TC (WDTC) [16], we performed analysis of the association of this hormone with ROM in patients with AUS/FLUS thyroid nodules. We found that women had a higher ROM than men, and the male/female ratio for TC is in agreement with that in other studies [2,28]. In our previous work, we evaluated selected ultrasound features of thyroid nodules assigned to the AUS/FLUS category to increase the sensitivity and specificity of this diagnosis [29]. We suggest that microcalcifications and rapid growth of the thyroid nodules might be used as predictive factors for TC development in patients with AUS/FLUS diagnosis. We also found that the risk of malignancy in AUS/FLUS depends upon specific clinical situations; however, we did not evaluate the hormonal status of these patients [29].

Some of the literature has evaluated the risk of TC according to TSH level in the normal range. In patients without thyroid hormonal dysfunction, a higher risk of malignancy was connected to TSH level between 1.0 and 1.7 mU/L compared to individuals with lower TC risk and TSH level below 0.4 mU/L, though the highest TC risk was observed in patients with TSH levels above 1.8 mU/L [30]. In another study, patients were divided according to TSH level, i.e., below 0.4 mU/L and above 3.4 mU/L, and patients in the first group had a significantly lower rate of papillary thyroid cancer than those in the second group (1.9 vs. 16.5%) [31].

Some authors have reported that a higher level of TSH predisposes older individuals toward an increased risk of TC and advanced tumor stage [32]. In our study, older patients (≥55 years old) with higher TSH levels had a higher risk of malignancy and advanced tumor stage than their younger counterparts.

We showed that nearly 14% of the thyroid nodules with AUS/FLUS diagnosis were malignant (47/342). Patients with thyroid nodules assigned to the third category of TBSRTC and with final histopathology showing malignancy had higher levels of TSH than those with benign tumors revealed by histopathological diagnosis.

Scintigraphy of thyroid nodule is still very important diagnostic tool, and should be performed in every cases with Bethesda III diagnosis. However, our analysis is retrospective, and we noticed the lack of this data, especially in the beginning of the analyzed period (2008–2015). In our analysis, in almost 90% of analyzed cases with histopathologic diagnosis of adenoma, we received this diagnosis after surgical resection. Almost all of these patients were qualified to surgery with presurgical diagnosis of thyroid tumor with III Bethesda category, and most of them were not qualified to scintigraphy. Indeed, some of them had a scintigraphy test performed, but only a few of them were described as a hot nodule.

In the study of Al Dawish et al. [2], the authors observed higher percentage values for ROM in patients with AUS/FLUS and TSH levels of >4.5 mIU/L. However, in some other studies, no relationship between TSH level and risk of malignancy was found [33,34]. Xiang et al. [15] performed a study to explore the relationship between TSH level and postoperative recurrence and lymph node metastasis in patients with papillary thyroid cancer (PTC), with patients with PTC and high levels of TSH (>2.615 mIU/L) having worse disease-free survival.

Baser et al. [16] conducted an interesting study in which they analyzed 1433 patients with various cytological diagnoses, including the AUS/FLUS category. The authors found that those with the V and VI categories of TBSRTC (suspicious for malignancy and malignant, respectively) had higher TSH levels than those with the other four categories (I–IV categories). The benign cytology group had significantly lower TSH levels than the other cytology groups, and TSH levels increased from Bethesda category II to VI [16]. The authors suggested that in addition to cytology, a higher TSH level can be used as a supplementary marker in the prediction of certain Bethesda categories.

Many authors indicate that AUS/FLUS patients comprise a group with ongoing challenges in terms of clinical management [5,6]. Although many patients with AUS/FLUS diagnosis qualify for repeated biopsy or even diagnostic surgery, many studies have focused on supportive molecular tests to enhance appropriate clinical management. However, such a strategy increases the complexity and cost of patient management. Thus, in recent years, a personalized approach in such cases has been recommended. To make the best clinical decision, all available information, such as TSH level, ultrasound features, and demographic details, should be analyzed. Hong et al. [35] suggested that some clinical factors should be assessed to establish ROM in patients with AUS/FLUS diagnosis. The necessity for subsequent analyses is also important because of the wide range of ROMs in resected AUS/FLUS nodules [2]. ROM in surgically resected nodules with AUS/FLUS category ranges from 6% to even 48% [2]. In our opinion, further investigations of clinical, biochemical and ultrasound factors of patients with AUS/FLUS diagnosis may improve the assessment of ROM and thus target treatment appropriately.

In contrast to many studies, Castro et al. [36] showed that in patients with thyroid nodules suspected by cytology, a higher ROM was associated with the presence of multiple nodules of smaller size. Kaya et al. [37] divided patients with Bethesda category III nodules according to the level of TSH serum concentration: below 1 mIU/L, 1–5 mIU/L, and above 5 mIU/L. Next, they analyzed the histopathology results in each group but did not observe any statistically significant differences (*p* = 0.321). They confirmed only hypoechoic structures, microcalcifications and irregular margins to be associated with malignancy in patients with FNAB results of AUS/FLUS.

Our work has some limitations. It was a retrospective study, and access to some necessary specific details was limited. Second, this study was observational, which makes it difficult to control for all potential confounding factors. Third, there was selection bias because the patients included in this study were admitted to the hospital, which indicates that they are not representative of the whole population. Fourth, as the analyzed data were from a single medical center, the possibility of selection bias cannot be ruled out. Fifth, a limited number of patients was analyzed. Sixth, no molecular tests were performed as an additional diagnostic tool in the prediction of ROM in AUS/FLUS patients.

In conclusion, we show an association between the AUS/FLUS Bethesda category and TSH level. Patients with thyroid nodules assigned to the AUS/FLUS category who had malignant final histopathology had higher TSH levels than those assigned to the same cytology but with benign final histopathology. In general, preoperative serum TSH level might be a useful factor for predicting TC in patients with AUS/FLUS thyroid tumor diagnosis. The combination of two risk factors for TC, i.e., AUS/FLUS diagnosis and higher TSH levels, may increase the ROM of TC.

## Figures and Tables

**Figure 1 biomedicines-10-01916-f001:**
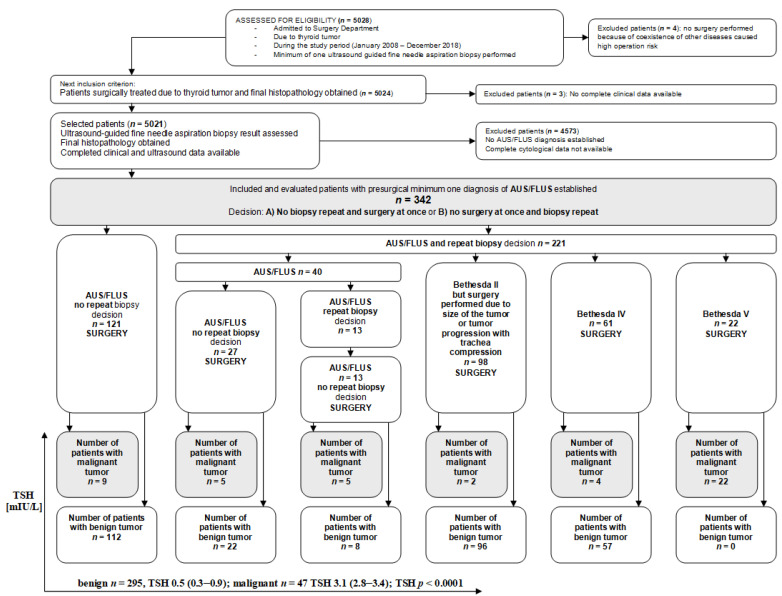
Selection of the study group from 5028 individuals referred for surgery from 2008 to 2018. All participants underwent a minimum of one UG-FNAB with a minimum of one AUS/FLUS diagnosis. All evaluated patients underwent surgery, and histopathology results were obtained in all cases. TSH serum levels were assessed before surgery in all patients.

**Figure 2 biomedicines-10-01916-f002:**
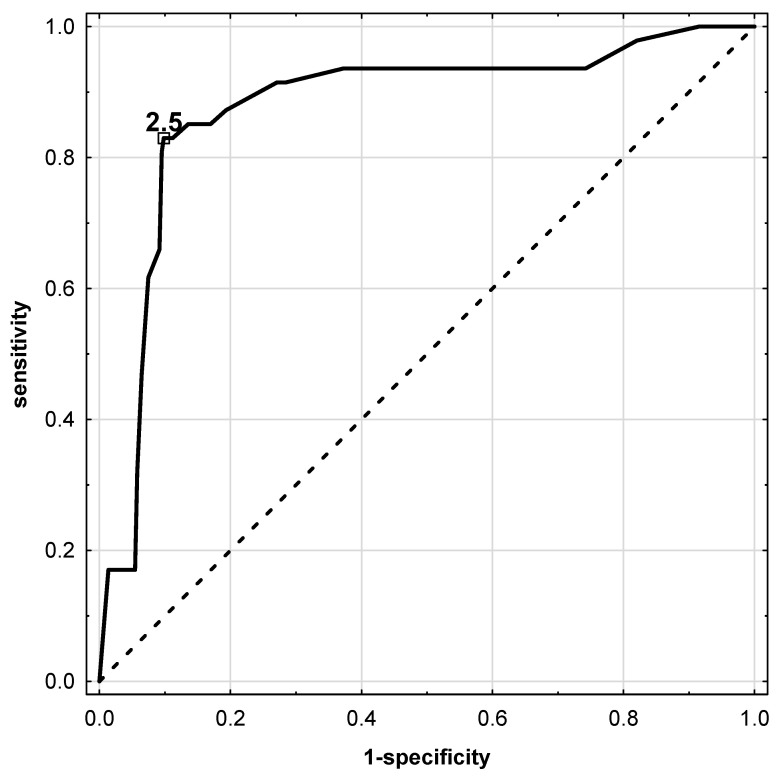
ROC analysis for TSH concentration in serum of patients before surgical treatment. Solid line—TSH parameter. Analysis showed cut-off point for prediction of thyroid cancer as ≥2.50 mU/L.

**Table 1 biomedicines-10-01916-t001:** Demographic, clinical, and histopathological characteristics of patients with AUS/FLUS diagnosis and final histopathological diagnosis of cancer. Descriptive data are presented as the number of observations (percent) or the mean ± SD.

Parameters		Total Group (*n* = 47)
Sex	Female	32 (68.1)
	Male	15 (31.9)
Age (years)	48.8 ± 15.4	
Age (years old)	<55 years old	37 (78.7)
	≥55 years old	10 (21.3)
Presurgical diagnosis	AUS	26 (12.1)
	FLUS	21 (87.9
Histopathological diagnosis	PTC	46 (97.9)
	FTC	1 (2.1)
Nodule size (cm)	≤1.0 cm	2 (4.2)
	>1.0 ≤ 2.0 cm	9 (19.2)
	>2.0 ≤ 4.0 cm	15 (31.9)
	>4.0 cm	21 (44.7)
pTNM:	I	11 (23.4)
	II	36 (76.6)
	III	-
	IV	-

AUS—atypia of undetermined significance; FLUS—follicular lesion of undetermined significance; PTC—papillary thyroid cancer; FTC—follicular thyroid cancer.

**Table 2 biomedicines-10-01916-t002:** Comparison of demographic and clinical parameters and ultrasound features between patients with histologically confirmed benign and malignant nodules. Descriptive data are presented as the number of observations (percent), mean (± SD) or median (Q1–Q3).

Parameters		Total Group (*n* = 342)	Benign (*n* = 295)	Cancer (*n* = 47)	*p* Value
Sex	Female Male	284 (83.0) 58 (17.0)	252 (85.4) 43 (14.6)	32 (68.1) 15 (31.9)	0.003 *
Age (years old)		51.3 ± 15.4	52.1 ± 15.5	46.0 ± 14.0	0.011 *
Age	<55 years old ≥55 years old	191 (55.9) 151 (44.2)	154 (52.2) 141 (47.8)	37 (78.7) 10 (21.3)	0.0007 *
Microcalcifications	Yes No	72 (21.1) 270 (78.9)	37 (12.5) 258 (87.5)	35 (74.5) 12 (25.5)	<0.0001 *
Echogenicity	Hypoechoic Hyperechoic	125 (36.6) 217 (63.4)	82 (27.8) 213 (72.2)	43 (91.5) 4 (8.5)	<0.0001 *
Irregular margin	Yes No	132 (38.6) 210 (61.4)	88 (29.8) 207 (70.2)	44 (93.6) 3 (6.4)	<0.0001 *
Taller than wide	Yes No	111 (32.5) 231 (67.5)	69 (23.4) 226 (76.6)	42 (89.4) 5 (10.6)	<0.0001 *
High vascularity	Yes No	118 (34.5) 224 (65.5)	88 (29.8) 207 (70.2)	30 (63.8) 17 (36.2)	<0.0001 *
Macrocalcifications	Yes No	158 (46.2) 184 (53.8)	142 (48.1) 153 (51.9)	16 (34.0) 31 (66.0)	0.072
TSH (mIU/L)		0.6 (0.4–1.5)	0.5 (0.3–0.9)	3.1 (2.8–3.4)	<0.0001 *

*—statistically significant; TSH—Thyroid-stimulating hormone.

**Table 3 biomedicines-10-01916-t003:** Diagnostic potential of TSH level as a marker of the presence of thyroid cancer.

AUC (±95% CI)	0.877 (0.818–0.935)
*p* value	<0.0001 *
sensitivity	0.830
specificity	0.902
accuracy	0.892
PPV	0.574
NPV	0.971
Youden index	0.731

*—statistically significant; 95%CI—95% confidence interval; AUC—area under the ROC curve; PPV—positive predictive value; NPV—negative predictive value.

**Table 4 biomedicines-10-01916-t004:** Univariable and multivariable logistic regression analysis of selected predictors of malignancy (benign/cancer; 0/1) in patients with AUS/FLUS thyroid tumor diagnosis.

Independent Variables	Univariable Analysis		Multivariable Analysis	
	OR (±95% CI)	*p*-Value	OR (±95% CI)	*p*-Value
Sex: for female	0.36 (0.18–0.73)	<0.004 *	2.36 (0.08–67.95)	0.615
Age: for <55 years old	3.38 (1.62–7.08)	0.001 *	108.9 (0.99–1189.62)	0.049 *
Microcalcifications	20.33 (9.67–42.76)	<0.0001 *	164.92 (2.62–1034.68)	0.015 *
Hypoechoic	27.92 (9.67–80.56)	<0.0001 *	250.97 (1.50–4186.21)	0.034 *
Irregular margin	34.49 (10.38–114.56)	<0.0001 *	12.69 (0.92–173.56)	0.056
Taller than wide	27.51 (10.43–72.51)	<0.0001 *	921.44 (2.36–3587.85)	0.024 *
High vascularity	4.15 (2.17–7.93)	<0.0001 *	19.46 (0.43–872.59)	0.124
TSH for ≥2.5mUI/L	40.26 (17.60–92.08)	<0.0001 *	168.69 (4.56–6237.33)	0.005 *

*—statistically significant; OR—odds ratio; 95%CI—95% confidence interval.

## Data Availability

The datasets used and/or analyzed during the current study are available from the corresponding author upon reasonable request.

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
