# Peer review of "Assessment of Preoperative TSH Serum Level and Thyroid Cancer Occurrence in Patients with AUS/FLUS Thyroid Nodule Diagnosis"

_biomedicines, 2022, doi:10.3390/biomedicines10081916_

Round 1

Reviewer 1 Report

This is a retrospective study that evaluates the blood TSH concentration in patients with Bethesda category III (AUS/FLUS) disease. The authors found that the TSH level above 2.5 mIU/L is a predictive for well-differentiated thyroid cancer. This is an interesting study, and the manuscript is well-written. Please address the following concerns.

1.  Table 3 is missing in the manuscript. It has been mentioned in page 6 line 1-3, however, the author did not provide the table.

“The importance of TSH levels, demographic parameters and ultrasound features as independent prognostic factors of thyroid malignancy were tested by univariable and multivariable logistic regression analyses (Table 3).”

 2.  ROC analysis showed the TSH at 2.5mIU/L is an optimal cutoff value with the AUC of 0.877. Although the author has provided the sensitivity/specificity/accuracy and PPV in Table 2, I think it would be better for the reader to understand if the author can show the figure of ROC curve

 3.  In this study, 342 patients with Bethesda category III were included. How many patients were diagnosed with AUS and how many patients were diagnosed with FLUS?

 4.  In this study, 47 patients were diagnosed with thyroid cancer by permanent section. Can you provide a summary of clinicopathological characteristics in cancer patients (e.g., tumor size, TNM stage, operation type…)?

 5.  In the result, 12.9%(n=44) patients had thyroiditis. Did you check ATA and anti-TPO before surgery? If one patient had normal TSH/FT4 but higher ATA, Anti-TPO, would he be enrolled in this study?

6.  page 4, bottom line, “The number of female patients with thyroid malignancy was significantly lower than that of male patients (p=0.003)” In this sentence, I think “lower” sould be” higher” please check again.

Author Response

Journal: Biomedicines                                                                       July 27, 2022

MDPI

Special Issue: Mechanism and Novel Therapeutic Approaches for Thyroid Diseases

Guest Editor: Professor Tommaso Aversa

Dear Editor and Reviewers,

At the very beginning we would like to thank you very much for the possibility to re-submit our revised manuscript entitled Assessment of Preoperative TSH Serum Level and Thyroid Cancer Occurrence in Patients with AUS/FLUS Thyroid Nodule Diagnosis.” for consideration as an original article in Special Issue Mechanism and Novel Therapeutic Approaches for Thyroid Diseases. Thank you very much for considering it for potential publication in Biomedicines.

We would like to thank you for the very thorough reviews and for the advices and constructive criticism, which have been valuable for improving our paper. All of the suggestions for changes and improvements were very helpful to us, and we have revised the manuscript according to the recommendations made in the reviews. All of the changed, deleted and added portions of the manuscript are marked by using Track Changes. According to the reviewers’ instructions we corrected our manuscript point-by-point as follows:

Reviewer 1

Dear Reviewer, thank you very much for your review and suggestions. Thank you very much for the statement, that “This is an interesting study, and the manuscript is well-written.”, it is wonderful and important opinion for us. Thank you.  As far as your suggestions, we introduced all of them to the manuscript as follows:

  1. Table 3 is missing in the manuscript. It has been mentioned in page 6 line 1-3, however, the author did not provide the table.

“The importance of TSH levels, demographic parameters and ultrasound features as independent prognostic factors of thyroid malignancy were tested by univariable and multivariable logistic regression analyses (Table 3).”

Ad. 1 Dear Reviewer, thank you very much for this remark. We are very sorry for this terrible omission we have made in our manuscript. We are very sorry for that. Honestly, we do not know how it happened. Maybe, because we were sending our manuscript two times to Nature Springer Author Service (certificate attached), we probably lost Table 3 during this process (or they?). We added Table 3 and the description of it to the manuscript. Because we formed another table (according to your advice, see below) the missing table has number 4. Thank you.

Table 4. Univariable and multivariable logistic regression analysis of selected predictors of malignancy (benign/cancer; 0/1) in patients with AUS/FLUS thyroid tumor diagnosis.

Independent variables

Univariable analysis

Multivariable analysis

OR (+ 95%CI)

p-value

OR (+ 95%CI)

p-value

Sex: for female

0.36 (0.18-0.73)

<0.004*

2.36 (0.08-67.95)

0.615

Age: for <55 years old

3.38 (1.62-7.08)

0.001*

108.9 (0.99-1189.62)

0.049*

Microcalcifications

20.33 (9.67-42.76)

<0.0001*

164.92 (2.62-1034.68)

0.015*

Hypoechoic

27.92 (9.67-80.56)

<0.0001*

250.97 (1.50-4186.21 )

0.034*

Irregular margin

34.49 (10.38-114.56)

<0.0001*

12.69 (0.92-173.56)

0.056

Taller than wide

27.51 (10.43-72.51)

<0.0001*

921.44 (2.36-3587.85)

0.024*

High vascularity

4.15 (2.17-7.93)

<0.0001*

19.46 (0.43-872.59)

0.124

TSH for >2.5mUI/L

40.26 (17.60-92.08)

<0.0001*

168.69 (4.56-6237.33)

0.005*

*: statistically significant; OR: odds ratio; 95%CI: 95% confidence interval

  1. ROC analysis showed the TSH at 2.5mIU/L is an optimal cutoff value with the AUC of 0.877. Although the author has provided the sensitivity/specificity/accuracy and PPV in Table 2, I think it would be better for the reader to understand if the author can show the figure of ROC curve

Ad. 2 Dear Reviewer, thank you so much for this suggestion. We added figure of ROC curve to the manuscript as follows. Thank you. 

Figure 2. ROC analysis for TSH concentration in serum of patients before surgical treatment. Solid line – TSH parameter. Analysis showed cut-off point for prediction of thyroid cancer as  >2.50 mU/L.

  1. In this study, 342 patients with Bethesda category III were included. How many patients were diagnosed with AUS and how many patients were diagnosed with FLUS?

Ad. 3 Dear Reviewer, thank you very much for this suggestion. Unfortunately, because it is retrospective study, and at the very beginning of the analyzed period (2008-2018), especially in the first part of the study (years 2008-2014) the cytologists often had been forming the diagnosis of “third category” of the Bethesda system, and they did not provide the distinguishing for AUS or FLUS, so we have only information of which category was assigned to, so in this occasion, we decided not to include the information for whole study group. However, we checked again all of the specimens from FNAB of the patients with final pathological diagnosis of cancer and we established the numbers of AUS and FLUS diagnoses of these patients, what we included in the table 1. We added also descriptive data. Thank you for this suggestion.  

Table 1. Demographic, clinical and histopathological characteristics of patients with AUS/FLUS diagnosis and final histopathological diagnosis of cancer. Descriptive data are presented as the number of observations (percent) or the mean + SD.

  1. In this study, 47 patients were diagnosed with thyroid cancer by permanent section. Can you provide a summary of clinicopathological characteristics in cancer patients (e.g., tumor size, TNM stage, operation type…)?

Ad. 4 Dear Reviewer, thank you very much for this suggestion. We formed the table containing some demographic, clinical and pathological characteristics of the patients with malignant tumor diagnosis. We included them in the table 1 as follows. Thank you very much for this suggestion.   

Parameters

Total group (n=47)

Sex

Female

32 (68.1)

Male

15 (31.9)

Age (years)

48.8 + 15.4

Age (years old)

<55 years old

37 (78.7)

>55 years old

10 (21.3)

Presurgical diagnosis

AUS

26 (12.1)

FLUS

21 (87.9

Histopathological diagnosis

PTC

46 (97.9)

FTC

1 (2.1)

Nodule size (cm)

≤1.0 cm

2 (4.2)

>1.0 ≤ 2.0 cm

9 (19.2)

>2.0 ≤ 4.0 cm

15 (31.9)

>4.0 cm

21 (44.7)

pTNM:

I

11 (23.4)

II

36 (76.6)

III

-

IV

-

AUS: atypia of undetermined significance; FLUS: follicular lesion of undetermined significance; PTC: papillary thyroid cancer; FTC: follicular thyroid cancer

  1. In the result, 12.9%(n=44) patients had thyroiditis. Did you check ATA and anti-TPO before surgery? If one patient had normal TSH/FT4 but higher ATA, Anti-TPO, would he be enrolled in this study?

Ad. 5 Dear Reviewer, thank you very much for this suggestion. Unfortunately, this is retrospective study, and in many analyzed cases, we did not find the data concerning antibodies levels before surgery. So in this occasion we did not performed the analysis of this variables. We included in the manuscript the information concerning potential limitations of the study:

“Our work has some limitations. It was a retrospective study, and access to some necessary specific details was limited.” We are very sorry for this occasion.

  1. page 4, bottom line, “The number of female patients with thyroid malignancy was significantly lower than that of male patients (p=0.003)” In this sentence, I think “lower” sould be” higher” please check again.

Ad. 6 Yes, thank you very much for this remark. We corrected this mistake as you suggested. Thank you.

Kind regards

The authors.

Reviewer 2 Report

I red with interest this paper that reports about the utility of TSH measurement in AUS/FLUS thyroid Nodules to predict final post-operative diagnosis of cancer.

The paper is interesting, well written and original.

I wonder why authors did not provide also data on Thyroglobulin measurements and compare its predictive value with pre-operative TSH.

A second point, is the large presence of cases of adenoma amongst the benign pathologies. Often, these adenomas presents low to very-low levels of TSH and may largely contribute to significance of results.

Since functioning adenomas can be easily diagnosed by 131I Thyroid scan, that should always be performed in case of an AUS/FLUS nodule, authors should analyze data also excluding the adenomas and verify if TSH levels are still predictive or not of malignancy.

Author Response

Journal: Biomedicines                                                                       July 27, 2022

MDPI

Special Issue: Mechanism and Novel Therapeutic Approaches for Thyroid Diseases

Guest Editor: Professor Tommaso Aversa

Dear Editor and Reviewers,

At the very beginning we would like to thank you very much for the possibility to re-submit our revised manuscript entitled Assessment of Preoperative TSH Serum Level and Thyroid Cancer Occurrence in Patients with AUS/FLUS Thyroid Nodule Diagnosis.” for consideration as an original article in Special Issue Mechanism and Novel Therapeutic Approaches for Thyroid Diseases. Thank you very much for considering it for potential publication in Biomedicines.

We would like to thank you for the very thorough reviews and for the advices and constructive criticism, which have been valuable for improving our paper. All of the suggestions for changes and improvements were very helpful to us, and we have revised the manuscript according to the recommendations made in the reviews. All of the changed, deleted and added portions of the manuscript are marked by using Track Changes. According to the reviewers’ instructions we corrected our manuscript point-by-point as follows:

Reviewer 2

Dear Reviewer, thank you very much for your review and suggestions. Thank you very much for the statement, that “the paper is interesting, well written and original”, it is wonderful and important opinion for us. Thank you.  As far as your suggestions, we introduced all of them to the manuscript as follows:

  1. I wonder why authors did not provide also data on Thyroglobulin measurements and compare its predictive value with pre-operative TSH.

Ad. 1 Dear Reviewer, thank you very much for this suggestion. Unfortunately, this is a retrospective study, and we do not have the data of thyroglobulin measurements  in all analyzed patients. We are very sorry for this. Because of the lack of this information in many patients we did not decide to include this analysis. We included in the manuscript information concerning some potential limitations of the study: “Our work has some limitations. It was a retrospective study, and access to some necessary specific details was limited. Second, this study was observational, which makes it difficult to control for all potential confounding factors.”

We formed additional table with some demographic, clinical and histopathological characteristics of all patients with malignant tumor diagnosis. We are very sorry for these missing data.

Table 1. Demographic, clinical and histopathological characteristics of patients with AUS/FLUS diagnosis and final histopathological diagnosis of cancer. Descriptive data are presented as the number of observations (percent) or the mean + SD.

Parameters

Total group (n=47)

Sex

Female

32 (68.1)

Male

15 (31.9)

Age (years)

48.8 + 15.4

Age (years old)

<55 years old

37 (78.7)

>55 years old

10 (21.3)

Presurgical diagnosis

AUS

26 (12.1)

FLUS

21 (87.9

Histopathological diagnosis

PTC

46 (97.9)

FTC

1 (2.1)

Nodule size (cm)

≤1.0 cm

2 (4.2)

>1.0 ≤ 2.0 cm

9 (19.2)

>2.0 ≤ 4.0 cm

15 (31.9)

>4.0 cm

21 (44.7)

pTNM:

I

11 (23.4)

II

36 (76.6)

III

-

IV

-

AUS: atypia of undetermined significance; FLUS: follicular lesion of undetermined significance; PTC: papillary thyroid cancer; FTC: follicular thyroid cancer

  1. A second point, is the large presence of cases of adenoma amongst the benign pathologies. Often, these adenomas presents low to very-low levels of TSH and may largely contribute to significance of results.

Ad. 2 Dear Reviewer, thank you very much for this remark. We checked the patients with final histopathologic diagnosis of adenoma, and we did not noticed significantly low levels of TSH in all analyzed cases. 

  1. Since functioning adenomas can be easily diagnosed by 131I Thyroid scan, that should always be performed in case of an AUS/FLUS nodule, authors should analyze data also excluding the adenomas and verify if TSH levels are still predictive or not of malignancy.

Ad. 3 Dear Reviewer, thank you very much for this suggestion. We absolutely agree, that scintigraphy of thyroid nodule is still very important diagnostic tool, and should be performed in every cases with Bethesda III diagnosis. However our analysis is retrospective, and we noticed lack of this data, especially in the beginning of the analyzed period (2008-2015). We included in the manuscript the statement of the limitation of the study. In our analysis, in almost 90% of analyzed cases with histopathologic diagnosis of adenoma, we received this diagnosis after surgical resection. Almost all of these patients were qualified to surgery with presurgical diagnosis “thyroid tumor with III Bethesda category”, and most of them were not qualified to scintigraphy. Indeed, some of them had scintigraphy test performed, but only few of them were described as a hot nodule. Thank you very much for your comments. We know, and we are very sorry that we could not include the data that you asked. 

Kind regards,

The authors.

Round 2

Reviewer 2 Report

Since authors agreed with my initial comments, I kindly ask the to include in the discussion the following sentence: "scintigraphy of thyroid nodule is still very important diagnostic tool, and should be performed in every cases with Bethesda III diagnosis. However our analysis is retrospective, and we noticed lack of this data, especially in the beginning of the analyzed period (2008-2015). In our analysis, in almost 90% of analyzed cases with histopathological diagnosis of adenoma, we received this diagnosis after surgical resection. Almost all of these patients were qualified to surgery with pre-surgical diagnosis “thyroid tumor with III Bethesda category”, and most of them did not perform thyroid scintigraphy. Indeed, some of them had scintigraphy test performed, but only few of them were described as a hot nodule".

Author Response

Journal: Biomedicines                                                                    August 1, 2022

MDPI

Special Issue: Mechanism and Novel Therapeutic Approaches for Thyroid Diseases

Guest Editor: Professor Tommaso Aversa

Dear Editor and Reviewers,

At the very beginning we would like to thank you very much for the second chance to re-submit our revised manuscript entitled Assessment of Preoperative TSH Serum Level and Thyroid Cancer Occurrence in Patients with AUS/FLUS Thyroid Nodule Diagnosis.” for consideration as an original article in Special Issue Mechanism and Novel Therapeutic Approaches for Thyroid Diseases. Thank you very much for considering it for potential publication in Biomedicines.

The added portion of the manuscript is marked by using Track Changes, and we corrected our manuscript point-by-point as follows:

Reviewer 2

Dear Reviewer, thank you very much for this suggestion. We added the statement, that “Scintigraphy of thyroid nodule is still very important diagnostic tool, and should be performed in every cases with Bethesda III diagnosis. However our analysis is retrospective, and we noticed lack of this data, especially in the beginning of the analyzed period (2008-2015). In our analysis, in almost 90% of analyzed cases with histopathologic diagnosis of adenoma, we received this diagnosis after surgical resection. Almost all of these patients were qualified to surgery with presurgical diagnosis of hyroid tumor with III Bethesda category, and most of them were not qualified to scintigraphy. Indeed, some of them had scintigraphy test performed, but only few of them were described as a hot nodule.” in discussion section of the manuscript. Thank you very much for your comment. Thank you.

Kind regards,

The authors.